# Exploration via Planning for Information about the Optimal Trajectory

**Viraj Mehta**[1], **Ian Char**[2], **Joseph Abbate**[4], **Rory Conlin**[4], **Mark D. Boyer**[3], **Stefano Ermon**[5], **Jeff Schneider**[1], **Willie Neiswanger**[5]

[1]Robotics Institute, [2]Machine Learning Department, Carnegie Mellon University
[3]Princeton Plasma Physics Laboratory, [4]Princeton University
[5]Computer Science Department, Stanford University

{virajm,ichar,schneide}@cs.cmu.edu, {jabbate,wconlin}@princeton.edu,
mboyer@pppl.gov, {ermon,neiswanger}@cs.stanford.edu

## Abstract

Many potential applications of reinforcement learning (RL) are stymied by the large numbers of samples required to learn an effective policy. This is especially true when applying RL to real-world control tasks, e.g. in the sciences or robotics, where executing a policy in the environment is costly. In popular RL algorithms, agents typically explore either by adding stochasticity to a reward-maximizing policy or by attempting to gather maximal information about environment dynamics without taking the given task into account. In this work, we develop a method that allows us to plan for exploration while taking both the task and the current knowledge about the dynamics into account. The key insight to our approach is to plan an action sequence that maximizes the expected information gain about the optimal trajectory for the task at hand. We demonstrate that our method learns strong policies with 2x fewer samples than strong exploration baselines and 200x fewer samples than model free methods on a diverse set of low-to-medium dimensional control tasks in both the open-loop and closed-loop control settings.[1]

## 1 Introduction

The potential of reinforcement learning (RL) as a general-purpose method of learning solutions to sequential decision making problems is difficult to overstate. Ideally, RL could allow for agents that learn to accomplish all manner of tasks solely through a given reward function and the agent's experience; however, RL has so far broadly fallen short of this. Chief among the difficulties in realizing this potential is the fact that typical RL methods in continuous problems require very large numbers of samples to achieve a near-optimal policy. For many interesting applications of RL this problem is exacerbated by the fact that collecting samples from the true environment can incur huge costs. For example, expensive scientific experiments are needed for tokamak control [31, 13, 20] and design of molecules [70], and collecting experience on the road is both costly and runs the risk of car accidents for autonomous vehicles [66].

Though model-based methods like Deisenroth and Rasmussen [21], Chua et al. [15], and Curi et al. [16] are much more efficient than typical model-free methods, they do not explicitly reason about the prospective task-relevant information content of future observations. Moreover, these methods typically require thousands of timesteps of environment dynamics data to solve reasonably simple Markov decision processes (MDPs). Though progress has been made in using Thompson sampling [46], entropy bonuses [27], and upper confidence bounds (UCB) [16, 5] to more intelligently explore

---

[1]Code is available at: https://github.com/fusion-ml/trajectory-information-rl

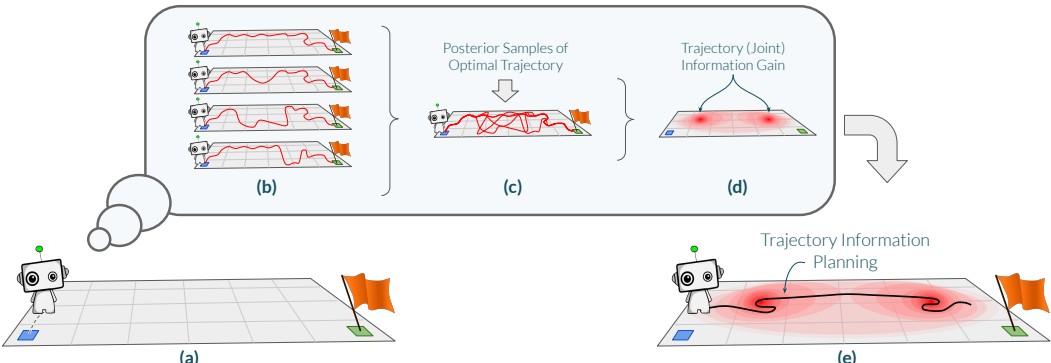

Figure 1: A schematic depiction of Trajectory Information Planning (TIP). Suppose the agent in (a) aims to determine where to explore next from its current state. To do so, in (b) the agent samples dynamics models $T' \sim P(T \mid D)$ from its current posterior and finds approximately optimal trajectories $\tau^* \sim P(\tau^* \mid T')$ for each sample. Then in (c) it pools these samples of posterior optimal trajectories $\tau^*$. In (d) it constructs a function that gives the joint expected information gain about the optimal trajectory $\tau^*$ given a planned exploration trajectory (i.e. $\text{EIG}_{\tau^*}$ over the set of points visited). Finally, in (e) the agent can plan an action sequence which maximizes this joint expected information gain.

the state-action space of an MDP, these methods still do not explicitly reason about how information that the agent gathers will affect the estimated MDP solution. Furthermore, in continuous state-action settings these methods must make coarse approximations to be computationally tractable (e.g. bootstrapped Q networks to approxiate a posterior or one-step perturbations for approximate UCB).

Many methods in the vein of Pathak et al. [47] and Shyam et al. [61] explore directly to gather new information based on current uncertainty about environment dynamics, but they do not in general specialize the information that they aim to acquire for a particular *task* specified by an initial state distribution and reward function. We believe that an ideal exploration strategy for sample efficiency should take into account this task, as well as uncertainty about the environment dynamics.

In this work, we start by showing how many methods can be cast as a Bayesian planning problem over a specific cost function. This framework helps illuminate the importance of the cost function and what impact it has on exploration. Viewed in this light, it becomes clear that many previous state-of-the-art methods rely on cost functions that either result in behavior that is too greedy—i.e. the policy tries to maximize returns during exploration—or too exploratory, i.e. the policy is incentivized to explore the environment dynamics and does not consider the task at hand. We therefore present a cost function that balances out these two extremes, by generalizing an information-gain acquisition function introduced in Mehta et al. [40] to apply to a set of future hypothetically acquired data. In particular, our cost function captures the amount of information that would be gained about the *optimal trajectory*, if the agent were to explore by following a particular planned trajectory. As depicted in Figure 1, this involves the agent sampling what it would hypothetically do given different realizations of the dynamics, and then planning actions that are informative about those possibilities.

In summary, the contributions of this work are as follows: we develop a novel cost function for exploration that explicitly accounts for both the specific task and uncertainty about environment dynamics, a method for planning which applies the cost function to explore in MDPs with continuous states and actions, and a thorough empirical evaluation of our method across 5 closed-loop and 3 open-loop environments (with a focus on expensive RL tasks in plasma physics) compared against 14 baselines. We find that our proposed method is able to learn policies that perform as well as an agent with access to the ground truth dynamics using half or fewer samples than comparison methods.

## 2  Related Work

**Exploration in Reinforcement Learning**   The most common strategy for exploration in RL is to execute a greedy policy with some form of added stochasticity. The simplest approach, $\epsilon$-greedy exploration as used in Mnih et al. [41], takes the current action thought to be best with probability $1 - \epsilon$ and a random action with probability $\epsilon$. Other methods use added Ornstein-Uhlenbeck action noise [37] to the greedy policy, or entropy bonuses [27] to the policy or value function objectives to add noise to a policy which is otherwise optimizing the RL objective.

Tabular RL is often solved by choosing actions based on upper confidence bounds on the value function [14, 36], but explicitly computing and optimizing these bounds in the continuous setting is substantially more challenging. Recent work [16] approximates this method by computing one-step confidence bounds on the dynamics and training a 'hallucinated' policy which chooses perturbations within these bound to maximize expected policy performance. Another recent work [5] uses anti-concentration inequalities to approximate upper confidence bounds in MDPs with discrete actions.

Thompson sampling (TS) [55], which samples a realization of the MDP from the posterior and acts optimally as if the realization was the true model, can be applied for exploration in a model-free manner as in [45] or in a model-based manner as in [63]. As the posterior over MDP dynamics or value functions can be high-dimensional and difficult to represent, the performance of TS can be hindered by approximation errors using both Gaussian processes and ensembles of neural networks. Curi et al. [16] recently investigated this and found that this was potentially due to an insufficiently expressive posterior over entire transition functions, implying that it may be quite difficult to solve tasks using sampled models. Similarly, the posterior over action-value functions in Osband et al. [45] is only roughly approximated by training a bootstrapped ensemble of neural networks.

There is also a rich literature of Bayesian methods for exploration, which are typically computationally expensive and hard to use, though they have attractive theoretical properties. These methods build upon the fundamental idea of the Bayes-adaptive MDP [53], which we detail in Section E.1 alongside a discussion of this literature.

Additionally, a broad set of methods explore to learn about the environment without addressing a specified task. This line of work is characterized by Pathak et al. [47], which synthesizes a task-agnostic reward function from model errors. Other techniques include MAX [61], which optimizes the information gain about the environment dynamics, Random Network Distillation [11], which forces the agent to learn about a random neural network across the state space, and Plan2Explore [60], which prospectively plans to find areas of novelty where the dynamics are uncertain.

**Bayesian Experimental Design: BOED, BO, BAX, and BARL**  There is a large literature on Bayesian optimal experiment design (BOED) [12] which focuses on efficiently querying a process or function to get maximal information about some quantity of interest. When the quantity of interest is the location of a function optimum, related strategies have been proposed as the entropy search family of Bayesian optimization (BO) algorithms [29, 30]. Recently, a flexible framework known as Bayesian algorithm execution (BAX) [43] has been proposed to efficiently estimate properties of expensive black-box functions; this framework gives a general procedure for sampling points which are informative about the future execution of a given algorithm that computes the property of interest, thereby allowing the function property to be estimated with far less data.

A subsequent related work [40], known as Bayesian Active Reinforcement Learning (BARL), uses ideas from BOED and BAX to sample points that are maximally informative about the optimal trajectory in an MDP. However, BARL relies on a setting the authors call Transition Query Reinforcement Learning (TQRL), which assumes that the environment dynamics can be iteratively queried at an arbitrary sequence of state-action pairs chosen by the agent. TQRL is thus a highly restrictive setting which is not suitable when data can only be accessed via a trajectory (rollout) of environment dynamics; it typically relies on an accurate environment simulator of sufficient expense to warrant its use. Even then, there will likely be differences between simulators and ground truth dynamics for complex systems. Thus, one would ideally like to collect data in real environments. However, this often requires leaving the TQRL setting, and instead collecting data via trajectories only.

In this paper, we aim to apply the information-theoretic ideas from BARL but generalize them to the general MDP setting as well as learn open loop model-based controllers. The typical method for learning to solve open-loop control problems was demonstrated successfully in Tesch et al. [65], where a value function was learned from action sequences to task success. Our method takes a model-based approach to this problem, using similar exploration strategies as Bayesian optimization but benefitting from the more substantial supervision that is typical in dynamics model learning.

## 3   Problem Setting

In this work we deal with finite-horizon discrete-time *Markov decision processes* (MDPs) which consist of a sextuple $\langle \mathcal{S}, \mathcal{A}, T, r, p_0, H \rangle$ where $\mathcal{S}$ is the state space, $\mathcal{A}$ is the action space, $T$ is the transition function $T : \mathcal{S} \times \mathcal{A} \to P(\mathcal{S})$ (using the convention that $P(\mathcal{X})$ is the set of probability measures over $\mathcal{X}$), $r : \mathcal{S} \times \mathcal{A} \times \mathcal{S} \to \mathbb{R}$ is a reward function, $p_0(s)$ is a distribution over $\mathcal{S}$ of start

states, and $H \in \mathbb{N}$ is the horizon (i.e. the length of an episode). We always assume $\mathcal{S}, \mathcal{A}, p_0$, and $H$ are known. We also assume the reward $r$ is known, though our development of the method can easily be generalized to the case where $r$ is unknown. Our primary object of interest is the transition function $T$, which we learn from data. We address both open and closed loop control settings. In the more common closed loop setting, our aim is to find a policy $\pi : \mathcal{S} \to \mathcal{A}$ that maximizes Objective (1) below. We will denote trajectories as $\tau \sim p(\tau \mid \pi, T)$ where $\tau = [(s_0, a_0), \ldots, (s_{H-1}, a_{H-1}), s_H]$ generated by $s_0 \sim p_0$, $a_i = \pi(s_i)$, and $s_{i+1} \sim T(s_i, a_i)$. We can write the return of a trajectory as $R(\tau) = \sum_{i=0}^{H-1} r(s_i, a_i, s_{i+1})$ for the states and actions $s_i, a_i$ that make up $\tau$. The MDP objective can then be written as

$$J_T(\pi) = \mathbb{E}_{\tau \sim p(\tau|\pi,T)} \left[ R(\tau) \right]. \tag{1}$$

We aim to maximize this objective while minimizing the number of samples from the ground truth transition function $T$ that are required to reach good performance. We denote the optimal policy as $\pi^* = \arg\max_\pi J_T(\pi)$, which we can assume to be deterministic [64] but not necessarily unique. We use $\tau^*$ to denote optimal trajectories, i.e. $\tau^* \sim p(\tau \mid \pi^*, T)$.

Similarly, for the open-loop setting, we assume a fixed start state $s_0$ and aim to find an action sequence $a_0, \ldots, a_{H-1}$ that maximizes the sum of rewards in an episode. We will slightly abuse notation and write $\tau \sim p(\tau \mid a_{0:H-1})$ and $J_T(a_{0:H-1})$ with these actions fixed in place of a reactive policy, and again use $\tau^*$ to refer to the trajectories generated by an optimal action sequence.

We assume in this work that applying planning algorithms like [51] to a dynamics function $T$ will result in a trajectory that approximates $\tau^*$. We will primarily focus on a Gaussian process (GP) model of the transition dynamics in order to take advantage of its expressive representation of uncertainty and grounded methods for sampling, conditioning, and joint inference. There is substantial prior work using GPs in RL—see Section E.2 for a discussion of this literature. Under this modeling choice, we assume that the dynamics are drawn from a GP prior $P(T)$ (see Section A.3 for further details on our GP model) and use $P(T \mid D)$ for the posterior over transition functions given a dynamics dataset of triples $D = \{(s_i, a_i, s_i')\}$. In this work, unions $D \cup \tau$ or $D \cup S'$ between the dataset $D$ and trajectories $\tau$ or next state predictions $S'$ coerce $\tau$ and $S'$ into triples of dynamics data, prior to the union with the dataset.

## 4 Trajectory Information Planning

Our method consists of a generic framework for Bayesian (or approximately Bayesian) model-predictive control and a novel cost function for planning that allows us to explicitly plan to find the maximal amount of new information relevant to our task. In Section 4.1, we describe the MPC framework and highlight that many prior methods approximate this framework while using a greedy cost function that corresponds to the future negative expected rewards or a pure exploration cost function that corresponds to future information about the dynamics. Afterwards, in Section 4.2, we derive our new cost function and describe how it is computed. The overall method we introduce simply applies this planning framework with our new cost function.

### 4.1 Model-Predictive Control in Bayesian Model-Based RL

In this section, we give a formulation of Bayesian planning for control that generalizes ideas from methods such as PILCO [21] and PETS [15]. This formulation highlights these methods' inherently greedy nature and hints at a possible solution. The objective of Bayesian planning is to find the $h$-step action sequence that maximizes the expected future returns under model uncertainty. That is,

$$\underset{a_0,\ldots,a_{h-1}}{\operatorname{argmin}} \ \mathbb{E}_{T' \sim P(T|D), \tau_e \sim P(\tau|s_0=s,a_{0:h-1},T')} \left[ C(\tau_e) \right] \tag{2}$$

for some cost function $C$ over trajectories and some start state $s$. If operating in the open-loop control setting, the agent executes the sequence of actions found without replanning. This procedure can also be extended to closed-loop control via model-predictive control (MPC), which involves re-planning (2) at every state the agent visits and playing the first action from the optimal sequence. Concretely, the MPC policy for our Bayesian setting is as follows:

$$\pi_{\text{MPC}}(s) = \underset{a_0}{\arg\min} \ \underset{a_1,\ldots,a_{h-1}}{\min} \ \mathbb{E}_{T' \sim P(T|D), \tau_e \sim P(\tau|s_0=s,a_{0:h-1},T')} \left[ C(\tau_e) \right] \tag{3}$$

Whether we do open-loop control or closed-loop control via MPC, the cost function $C$, is integral to how the agent will behave. Prior work has predominantly focused on two types of cost function:

$$\underbrace{C_g(\tau) = -R(\tau)}_{\text{Greedy Exploration}} \qquad \underbrace{C_e(\tau) = -\sum_{i=0}^{h} \mathbb{H}\left[T(s_i, a_i) \mid D\right]}_{\text{Task-Agnostic Exploration}} \qquad (4)$$

Previous works such as Kamthe and Deisenroth [34] and PETS [15] use the greedy exploration cost function, $C_g$. This cost function incentivizes trajectories that achieve high rewards over the next $h$ transitions on average. In works that focus on task-agnostic exploration such as Sekar et al. [60] and Shyam et al. [61], the cost function $C_e$ (or similar) is used to encourage the agent to find areas of the state space in which the model is maximally uncertain. Note that we use $\pi_g$ to refer to the greedy policy given by using (3) with $C_g$.

The optimization problem in (3) is typically approximately solved in one of three ways: Deisenroth and Rasmussen [21] and Curi et al. [16] directly backpropagate through the estimated dynamics and reward functions to find policy parameters that would generate good actions, Janner et al. [32] use an actor-critic method trained via rollouts in the model alongside the data collected to find a policy, and Chua et al. [15] and Mehta et al. [40] use the cross-entropy method [17] to find action sequences which directly maximize the reward over the estimated dynamics. In this work, we use a version of the last method given in Pinneri et al. [51], denoted iCEM, to directly find action sequences that optimize the cost function being used. We approximate the expectation by playing the actions on multiple samples from the posterior $P(T \mid D)$. Algorithm 1 gives a formal description of the method and Section A.5 provides further details.

---

**Algorithm 1** Bayesian Model-Predictive Control with Cost Function $C$

---

**Inputs:** transition function episode query budget $b$, number of posterior function samples $k$, planning horizon $h$.
Initialize $D \leftarrow \emptyset$.
**for** $i \in [1, \ldots, b]$ **do**
    Sample start state $s_0 \sim p_0$.
    **for** $t \in [0, \ldots, H-1]$ **do**
        Sample posterior functions $\{T'_\ell\}_{\ell=1}^{k} \sim P(T' \mid D)$.
        Approximately find $\arg\min_{a_0, \ldots, a_{h-1}} \sum_{\ell=1}^{k} \mathbb{E}_{\tau_\ell \sim p(\tau \mid T'_\ell, a_0, \ldots, a_{h-1})}\left[C(\tau_\ell)\right]$ via iCEM.
        Execute action $a_0$ by sampling $s_{t+1} \sim T(s_t, a_0)$.
        Update dataset $D \leftarrow D \cup \{(s_t, a_0, s_{t+1})\}$.
    **end for**
**end for**
**return** $\pi_g$ for the posterior $P(T' \mid D)$.

---

### 4.2 A Task-Specific Cost Function based on Trajectory Information

In this work, we aim to explore by choosing actions that maximize the conditional expected information gain (EIG) about the optimal trajectory $\tau^*$. This is the same overall goal as that of Mehta et al. [40], where the $\text{EIG}_{\tau^*}$ acquisition function was introduced for this purpose. However, in this paper we generalize this acquisition function in order to allow for sequential information collection, and account for the redundant information that could be collected between timesteps. As discussed at length in Osband et al. [46], it is essential to reason about how an action taken at the current timestep will affect the possibility of learning something useful in future timesteps. In other words, exploration must be *deep* and not greedy. Explicit examples are given in Osband et al. [46] where the time to find an $\epsilon$-optimal policy in a tabular MDP is exponential in the state size unless exploration can be coordinated over large numbers of timesteps rather than being conducted independently at each action. As the $\text{EIG}_{\tau^*}$ acquisition function is only defined over a single state-action pair and mutual information is submodular, we cannot naively use the acquisition function as is (or sum it over many datapoints) to choose actions that lead to good long-term exploration. This is clear in e.g. navigation tasks, where the nearby points visited over trajectories will provide redundant information about the local environment.

We therefore give a cost function that generalizes $\text{EIG}_{\tau^*}$ by taking a set of points to query and computing the *joint* expected information gain from observing the set. Our cost function is non-

Markovian in the state space of the MDP, but it is Markovian in the dataset, which represents a point in the belief space of the agent about the dynamics. Let $\mathcal{X} = \{x : x \subseteq \mathcal{S} \times \mathcal{A}, |x| < \infty\}$ be the set of finite subsets of the set of all state-action pairs. Our cost function $C_{\tau^*} : \mathcal{X} \to \mathbb{R}$ is defined below to be the negative *joint expected information gain* about the optimal trajectory $\tau^*$ for a subset $X \in \mathcal{X}$. In particular, assuming an existing dataset $D$, a set of $h$ query points $X = \{(s_i, a_i)\}_{i \in [h]}$, and a random set of next states $S' = \{s'_i \sim T(s_i, a_i), i \in [h]\}$,

$$C_{\tau^*}(X) = \mathbb{E}_{S' \sim p(S'|X,D)} \left[ \mathbb{H}\left[\tau^* \mid D \cup S'\right] \right] - \mathbb{H}\left[\tau^* \mid D\right]. \tag{5}$$

This formulation of $C_{\tau^*}$ forces our method to handle the redundant information among queries—it is likely that $I(s'_1, \tau^*) + I(s'_2, \tau^*) > I(\{s'_1, s'_2\}, \tau^*)$ and our method should avoid this overestimation. However, as written, this function relies on computing entropies on high-dimensional trajectories where the form of the joint distribution of the elements is unknown. To tractably estimate this quantity, we use the fact that $C_{\tau^*}(X) = -I(S', \tau^*) = -I(\tau^*, S')$ for the mutual information $I$. This allows us to exchange $\tau^*$ and our set of queries so that $\tau^*$ is giving information about the posterior predictive distribution of our set. In other words,

$$C_{\tau^*}(X) = \mathbb{E}_{\tau^* \sim p(\tau^*|D)} \left[ \mathbb{H}\left[S' \mid D \cup \tau^*\right] \right] - \mathbb{H}\left[S' \mid D\right]. \tag{6}$$

In order to compute the right-hand term, we must take samples $\tau^*_{ij} \sim P(\tau^* \mid D), i = 1, \ldots, m, j = 1, \ldots, n$. To do this, we first sample $m$ start states $s_0^{(i)}$ from $p_0$ (we always set $m = 1$ in experiments but derive the procedure in general) and for each start state independently sample $n$ posterior functions $T'_{ij} \sim P(T' \mid D)$ from our posterior over dynamics models. We then run a planning procedure using iCEM [51] on each of the posterior functions from $s_0^{(i)}$ using $T'_{ij}$ for $T$ (using our assumption that planning can generate approximately optimal trajectories given ground-truth dynamics), giving our sampled $\tau^*_{ij}$. Formally, we can approximate $C_{\tau^*}$ via Monte-Carlo as

$$C_{\tau^*}(X) \approx \frac{1}{mn} \left( \sum_{i \in [m]} \sum_{j \in [n]} \mathbb{H}[S'|D \cup \tau^*_{ij}] \right) - \mathbb{H}[S' \mid D]. \tag{7}$$

Assuming the dynamics are modelled with a Gaussian process, we can compute the joint Gaussian probability of the next states $S'$ [52]. As the entropy of a multivariate Gaussian depends only on the log-determinant of the covariance, $\log |\Sigma|$, we can tractably compute the joint entropy of the model predictions $\mathbb{H}\left[S' \mid D\right]$ and optimize it with a zeroth order optimization algorithm. Finally, we must calculate the entropy $\mathbb{H}[S'|D \cup \tau^*_{ij}]$. For this, we follow a similar strategy as Neiswanger et al. [43]: since $\tau^*_i$ is a set of states given by the transition model, we can treat them as additional noiseless datapoints for our dynamics model and condition on them before computing the joint covariance matrix for $S'$. Given this newly generalized acquisition function, we can instantiate a method of planning in order to maximize future information gained. We give the concrete procedure for computing our acquisition function in Algorithm 2, noting that trajectories $\tau^*_{ij}$ do not depend on the query set $X$ and can be cached for various values of $X$ as long as the dataset $D$ does not change.

Our ultimate procedure, which we name *Trajectory Information Planning* (TIP), is quite simple: run model-based RL using MPC as in Algorithm 1, but set the cost function to be $C_{\tau^*}(\tau)$ instead of $C_g$ or $C_e$, and compute this cost function using Algorithm 2. At test time, we return to planning with $C_g$ as the cost function and greedily attempt to maximize returns rather than performing exploration. We can also formulate an open-loop variant of our method, oTIP, which involves planning once and then executing the entire action sequence.

### 4.3 Computational Cost and Implementation Details

Though the TIP algorithm is designed for settings where samples are expensive, it is important to understand, both theoretically and practically, the computational cost of this method. For ease of notation, we make the simplifying assumption that the planning algorithm used (in this case, iCEM from [51]) evaluates $p$ action sequences consisting of $h$ (the planning horizon) actions and that our current dataset is of size $N$. In order to efficiently sample functions from the posterior over dynamics functions, we use the method from Wilson et al. [69]. This reduces the naive complexity of querying these functions from $O(N^3)$ to a one-time $O(N)$ cost and then $O(1)$ for additional queries. As we derive in Section A.1, the computational complexity of one TIP planning iteration is $O\left(nm\left((N+H)^3 + ph(N+H)^2\right)\right)$. The two asymptotically expensive operations are (1)

---
**Algorithm 2** Computation of $C_{\tau^*}$
---

**Inputs:** dataset $D = \{(s_k, a_k, s'_k)\}$, query set $X$, number of start state samples $m$, number of posterior function samples $n$.

Sample $m$ start states $\{s_0^{(i)}\}_{i=1}^m \sim p_0$.

**for** $i \in [m]$ **do**

    Sample $n$ posterior functions $\{T'_j\}_{j=1}^n \sim P(T' \mid D)$.

    **for** $j \in [n]$ **do**

        Set $\pi_j^* \leftarrow \pi_{\text{MPC}}$ using $C_g$ and a singleton posterior $P(T \mid D) = \delta(T'_j)$ as in (3).

        Compute $\tau_{ij}^*$ by executing $\pi_j^*$ on $T'_j$ starting from $s_0^{(i)}$.

    **end for**

**end for**

Compute the joint posterior covariance $\Sigma^{S'} \mid D$ across all points in $X$.

Compute the joint posterior covariances $\Sigma_{ij}^{S'} \mid D \cup \tau_{ij} \, \forall i \in [n], j \in [m]$ across all points in $X$.

**return** $\log |\Sigma^{S'}| - \frac{1}{nm} \sum_{i \in [n], j \in [m]} \log |\Sigma_{ij}^{S'}|$.

---

computing the Cholesky decompositions of the $nm$ kernel matrices for datasets $D \cup \tau_{ij}^*$ and (2) solving the triangular systems using the cached Cholesky decompositions in order to compute the covariance matrices $\Sigma_{ij}^{S'} \mid D \cup \tau_{ij}^*$ for each of the $p$ action sequences used by the planning algorithm.

However, our implementation choices mean that in practice these operations are not the most expensive step. The covariance matrix computations, which are the theoretical bottleneck, are implemented in JAX [9], allowing them to be compiled to much faster machine code and vectorized across large batches of queries. In fact, the most expensive operation in practice is planning on the sampled transition functions $T'_i$ to sample optimal trajectories $\tau_{ij}^*$. This is due to the fact that in practice $p$ is large and we implemented the planner in NumPy [28] so it cannot be compiled together with the Tensorflow [1] code from Wilson et al. [69], which is used for predicting which states will be visited for the planner. We give further information on the implementation in Section A.

## 5 Experiments

The aim of our development of the TIP algorithm and the $C_{\tau^*}$ acquisition function for RL is to reduce the sample complexity of learning effective policies in continuous MDPs given limited access to expensive dynamics. In this section we demonstrate the effectiveness of TIP in quickly learning a good policy by comparing against a variety of state-of-the-art reinforcement learning algorithms and strong baselines (including some that use the TQRL setting from [40], which is also known as RL with a generative model in Kakade [33] and other works [6, 4]).

In particular, we compare the average return across five evaluation episodes across five random seeds of each algorithm on five closed-loop control problems. For sample complexity we assess the median amount of data taken by each algorithm to 'solve' the problem across five seeds with the threshold performance given by an MPC controller using the ground truth dynamics. We evaluate the open-loop variant of our method, oTIP, against three comparison methods on three control problems suitable for open-loop control. In particular, to be suitable for open-loop control, the problem cannot be dynamically unstable (as Pendulum and Cartpole famously are) and must have a relatively short control horizon and fixed start state.

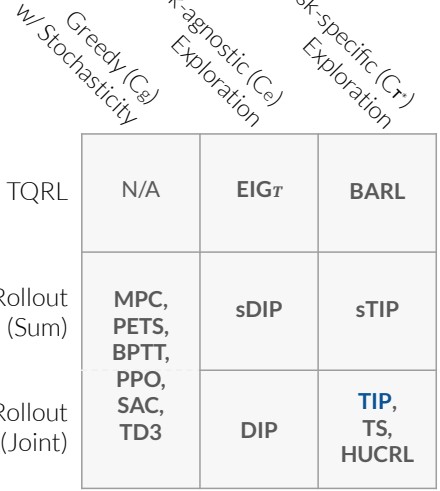

Figure 2: Our comparison methods can be broken down by the type of cost function used and how the methods do or do not handle sequential acquisition of information. As $C_g$ is a sum, it naturally handles future timesteps jointly. For the other information quantities, it is possible to upper-bound information acquired by summing each separate mutual information, or to compute them jointly.

| Environment | TIP | sTIP | DIP | MPC | PETS | SAC | FEEF | RHC | HUCRL | TS | BARL | $\mathrm{EIG}_T$ |
|---|---|---|---|---|---|---|---|---|---|---|---|---|
| Pendulum | **21** | 36 | 36 | 46 | 5.6k | 7k | 800 | >40k | >50k | >50k | **21** | 56 |
| Cartpole | 131 | 141 | 161 | 201 | 1.63k | 32k | >2.5k | >5k | >6k | >6k | **111** | 121 |
| $\beta$ Tracking | **46** | 76 | 276 | 76 | 330 | 12k | 300 | >3k | 480 | 420 | 186 | >1k |
| $\beta$ + Rotation | **201** | >500 | >500 | >500 | 400 | 30k | >2k | >2k | >5k | >5k | >500 | >1k |
| Reacher | **251** | >400 | >1k | 751 | 700 | 23k | >5k | 1.5k | 6.6k | 4.5k | **251** | >1.5k |

Table 1: **Sample Complexity:** Median number of samples across five seeds required to reach 'solved' performance, averaged across five trials. We determine 'solved' performance by running an MPC policy (similar to the one used for evaluation) on the ground truth dynamics to predict actions. We record $> n$ when the median run is unable to solve the problem by the end of training after collecting $n$ datapoints. The methods in the rightmost section operate in the TQRL setting and therefore have more flexible access to the MDP dynamics for data collection. The full set of methods are shown in Section D as well as boxplots depicting the data in Figure 4.

Here too, we assess the average return as open-loop trials are conducted as well as the number of timesteps required to achieve 'solved' performance.

**Comparison Methods**   We use several model-based and model-free comparison methods in this work. We compare to several published model-based methods. These include **PETS** [15], as implemented by Pineda et al. [50], which uses a probabilistic ensemble of neural networks and CEM over particle samples to do MPC. We also compare against three model-based techniques from the HUCRL [16] implementation: **HUCRL** itself, which relies on hallucinating dynamics perturbations as a way of realizing an upper confidence bound on the policy, model-based Thompson Sampling (**TS**), which samples from the posterior over models and chooses optimal actions for that sample, and a greedy model-based neural network method relying on backpropagation through time (**BPTT**). We also compare against the Free Energy of the Expected Future method from Tschantz et al. [67], which treats directed exploration as a process of actively collecting information for inference on a reward-biased generative model. A further comparison is with Receding Horizon Curiosity (RHC) [59], which does online Bayesian system identification over a linear model in order to quickly find a model of the environment dynamics. Our model-free comparison methods, Soft Actor-Critic (**SAC**) [27], an actor-critic method that uses an entropy bonus over the policy to encourage more exploration, and two others (TD3 and PPO), are in the appendix.

Finally, we compare against various ablations of the proposed method. These vary across two axes as described in Figure 2: the cost function they use and how they handle sequential queries. Besides these differences, they use the same GP model and iCEM planning algorithm with the same hyperparameters, so they are truly comparable methods. The three cost functions used are $C_g$, $C_{\tau^*}$, and $C_e$. **DIP**, **oDIP**, and **EIG**$_T$ all use $C_e$ but compute, respectively, the expected joint entropy of the action sequence, the sum of the pointwise entropies of the action sequence, and the individual pointwise entropies in the TQRL setting. These methods are very similar in spirit to [61, 48] in that they plan for future information gain about the dynamics, but we chose to compare in a way that controls for difference in the model and planning algorithm. **MPC** uses $C_g$ and is very close to the method in [34]. Like TIP, **BARL** [40] and **sTIP** use $C_{\tau^*}$. BARL operates in the TQRL setting and can therefore use the simpler $\mathrm{EIG}_{\tau^*}$ acquisition function. sTIP investigates the use of $-\sum_{s_i, a_i \in S} \mathrm{EIG}_{\tau^*}(s_i, a_i)$ as a cost function for planning. This computes individual information gains for each future observation without accounting for the information they may have in common and is therefore an overestimate of the joint information gain. In the open-loop setting we compare against **oDIP** and **oMPC**, the open-loop variants of DIP and MPC, and Bayesian optimization (**BO**) as implemented by Pedregosa et al. [49]. oDIP plans an action sequence to minimize the joint $C_e$ and executes the actions found for each open-loop trial, while oMPC does the same thing using $C_g$. We give additional details on the comparison methods in Section B.

**Control Problems**   Our closed loop control problems are the standard underactuated **Pendulum** swing-up task (Pendulum-v0 from Brockman et al. [10]) with 2D states and 1D actions, a **Cartpole** swing-up task with a sparse reward function, 2D s, and 1D actions, a 2-DOF robot arm control problem where the end effector is moved to a goal (**Reacher**-v2 from Brockman et al. [10]) with 10D states and 2D actions, a simplified $\beta$ **Tracking** problem from plasma control [13, 39] (similar in design but not identical to the one from Mehta et al. [40]) trained using with 4D states and 1D actions, and a more complicated problem in plasma control where $\beta$ **+ Rotation** are tracked with 10D states and 2D actions. Our open loop control problems are a navigation problem with hazards (**Lava Path**) from [40] and two regulation problems with different **Nonlinear Gain** functions. The Lava Path problem has 4D states and 2D actions and the nonlinear regulation problems have 2D states and 2D actions. Full details on these problems are available in Section C.

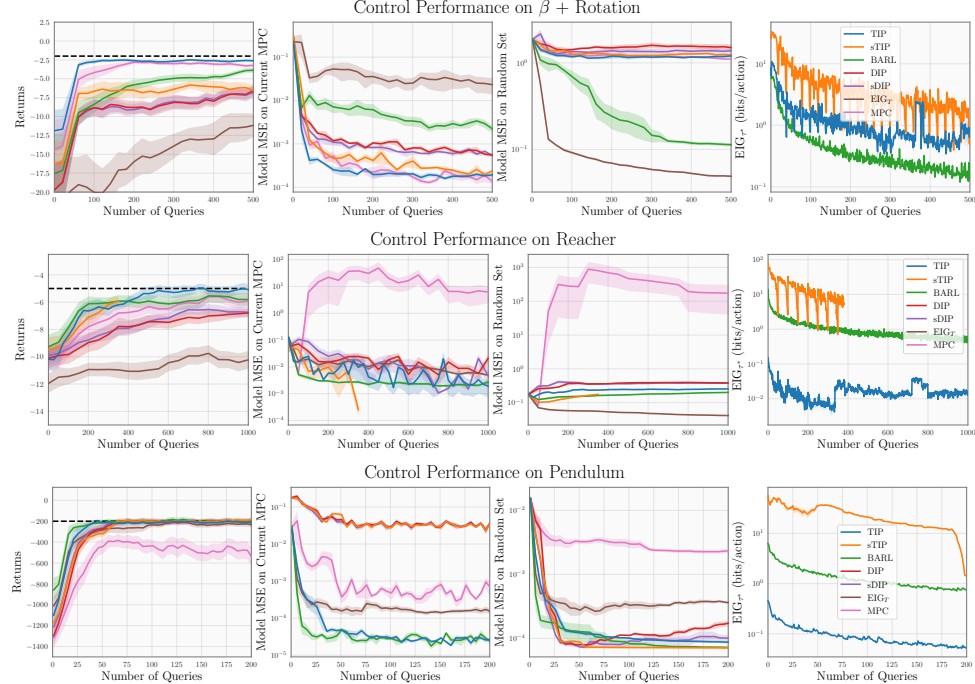

Figure 3: **Control and Modeling Details for TIP and Ablations.** Column 1: Learning curves for our ablation methods, all of which use the same planner and model. Column 2: Dynamics model accuracy on the points used by the planner to choose actions during MPC. Column 3: Dynamics model accuracy on a uniformly random test set in $\tilde{\mathcal{S}}$. Column 4: $\text{EIG}_{\tau^*}$ values normalized by the number of actions planned. sTIP was truncated on Reacher as it exceeded the wall time budget.

**Results** As can be seen in Table 1, TIP is able to reach solved performance more quickly across the board than the model-based and model-free external baselines, often using a fraction or even orders of magnitude less data than other methods. For many of our ablation methods we see failures to solve some of the problems even though the model is demonstrated by TIP to be able to sufficiently predict the dynamics. This is especially apparent on the harder plasma control environment, $\beta$+Rotation, where TIP is the only method using our GP which is able to solve the problem. We believe that this is

| Environment | oTIP | oMPC | oDIP | BO |
|---|---|---|---|---|
| Nonlinear Gain 1 | **41** | 91 | 51 | 210 |
| Nonlinear Gain 2 | **51** | 61 | >200 | 60 |
| Lava Path | **41** | 101 | 101 | >2k |

Table 2: **Open Loop Sample Complexity:** Median number of samples required to reach 'solved' performance, averaged across five trials. We determine 'solved' performance by running an MPC policy on the ground truth dynamics to predict actions. We record $> n$ when the median run is unable to solve the problem by the end of training after collecting $n$ datapoints.

because the data acquired through exploration by the ablation methods is less useful for control than the data TIP collects. This is underscored by the second column of Figure 3, where it is clear that TIP achieves the lowest modeling error on the points actually needed during the execution of the policy but not on the uniform test set. In particular we find it interesting that TIP outperforms BARL on the $\beta$ + Rotation environment, as BARL should in principle have a strictly stronger access to the problem and is optimizing the same quantity with fewer constraints. We hypothesis that this may be due to the fact that BARL optimizes the acquisition function $\text{EIG}_{\tau^*}$ by simply uniformly sampling a set of points and choosing the one that evaluates to the largest value. Our more sophisticated optimization algorithm and forced initialization at the start state distribution seems to allow us to collect more information in this case. This interpretation is bolstered by the fact that on the problems where TIP outperforms BARL, we see that TIP is actually collecting more information per action than BARL as evidenced by larger EIG values. We also see clearly that there is value in computing the $C_{\tau^*}$ function rather than summing over $\text{EIG}_{\tau^*}$ values, as TIP outperforms sTIP across the board. Additionally, there is clear evidence for the value of task-specific exploration as the task-agnostic exploration methods ($\text{EIG}_T$, DIP, sDIP) underperform both in returns and model error on the trajectories visited.

For the open-loop experiments (Table 2), we also see strong performance from oTIP. As the model-based methods benefit from observing many model transitions for each open-loop trial, it is unsurprising that they are more sample-efficient than the BO method. Within the model-based techniques, oTIP is the most sample efficient. We believe that this is for much the same reasons as in the closed-loop case—exciting evidence that the $C_{\tau^*}$ cost function can be applied in a variety of settings.

## 6   Conclusion

In this work, we presented and evaluated a cost function designed for intelligent, task-aware exploration. Using this cost function in model-predictive control allows agents to solve continuous MDPs with far less data than comparison methods in open- and closed-loop settings. Though the method is effective in data reduction, it is computationally expensive and relies on dynamics that are well-modeled by a GP. In future work, we aim to scale the method to higher-dimensional and more complex control environments. We also aim to apply this method in the real world. In particular, we aim to address similar plasma control problems in a small number of trials on a real tokamak.

### Acknowledgements

This work was supported in part by US Department of Energy grants under contract numbers DE-SC0021414 and DE-AC02-09CH1146.

This work is supported by the National Science Foundation Graduate Research Fellowship Program under Grant No. DGE1745016 and DGE2140739. Any opinions, findings, and conclusions or recommendations expressed in this material are those of the author(s) and do not necessarily reflect the views of the National Science Foundation.

This work is additionally supported by NSF (#1651565), AFOSR (FA95501910024), ARO (W911NF-21-1-0125), CZ Biohub, and Sloan Fellowship.

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
