# A  Implementation Details

## A.1  Derivation of Computational Cost

In this section, we derive the computational complexity of the TIP algorithm. For simplicity, we focus on a single TIP planning iteration as might be done at each replanning in closed-loop control or at the start of a trial in open-loop control. In order to keep the analysis general, we assume that the chosen planning algorithm requires $p$ accesses to the model where $h$ actions are sequentially executed, giving $ph$ total queries per planner execution. We also assume that the numbers of inducing points and basis functions used in our GP posterior function sampling are constant as are the Monte Carlo hyperparameters $m, n$.

The TIP algorithm consists of the following major operations:

- Sample $T_i'$ for $j \in [m]$: $O(nmN)$ total cost from sampling algorithm, where $N$ is the dataset size.

- Sample $\tau_{ij}^*$ for $i \in [n], j \in [m]$: $phHnm$ total cost from running the planner ($ph$ posterior function queries) $H$ times for each sampled $\tau^*$, where $H$ is the MDP horizon.

- Compute Cholesky decomposition for each $D \cup \tau_{ij}^*$. This takes a total of $O(nm(N + H)^3)$ operations as the augmented dataset is of size $N + H$ and Cholesky decompositions are $O(d^3)$ in the matrix size $d$.

- Compute posterior covariance $\Sigma^{S'} \mid D \cup \tau_{ij}$ for all $\tau_{ij}$. This involves several matrix operations but the most computationally intensive is solving $h$ triangular systems of size $(N + H) \times (N + H)$, which each take $O((N + H)^2)$ time. So the total computation here is $O(pnmh(N + H)^2)$.

- Compute determinants of covariance matrices for each of $p$ queries and $nm$ augmented datasets $D \cup \tau_{ij}$. Each of these operations is over a matrix of size $h \times h$ and therefore costs $O(h^3)$. So the total cost is $O(pnmh^3)$.

Summing these costs gives $O(nmN + phHnm + nm(N + H)^3 + pnmh(N + H)^2 + pnmh^3)$. Clearly the third term dominates the first, the fourth dominates the second, and since $H > h$, the fourth dominates the fifth. So, the computational cost can be summarized as $O\left(nm\left((N + H)^3 + ph(N + H)^2\right)\right)$.

## A.2  Wall Times

Though TIP and oTIP are designed for applications where samples are expensive and computation is relatively inexpensive, we present in this section data on the running time of these methods. We ran all experiments on a shared research cluster available to us on large machines with hundreds of GB of memory and between 24 and 88 CPU cores. In general our implementation did not make use of more than 20 CPU cores concurrently. In Table 3, we give the running time of the phases of the TIP algorithm. We note that the bulk of the computation in the planning procedure actually goes towards the just-in-time compilation of the JAX code that computes the cost function $C_{\tau^*}$ on sampled future trajectories. In order to allow for this compilation cost, we modified the iCEM algorithm from [51] to take fixed batch sizes as the compilation (e.g. for the $\beta$ tracking problem) takes approximately 90% of the time required for planning. Unfortunately this compilation process must be repeated at every iteration due to the limitations of the JAX compiler. We believe that a similarly JIT-compiled implementation of the planning algorithm for sampling $\tau^*$ on posterior samples could lead to a substantial speedup and a more flexible compiler could do more still.

## A.3  GP Model Details

For all of our experiments, we use a squared exponential kernel with automatic relevance determination [38, 42]. The parameters of the kernel were estimated by maximizing the likelihood of the parameters after marginalizing over the posterior GP [68].

To optimize the transition function, we simply sampled a set of points from the domain, evaluated the acquisition function, and chose the maximum of the set. This set was chosen uniformly for every problem but $\beta$ + Rotation and Reacher, for which we chose a random subset of $\cup_i \cup_j \tau_{ij}^*$ (the posterior

| Control Problem | Pendulum | Cartpole | $\beta$ Tracking | $\beta$ + Rotation | Reacher |
|---|---|---|---|---|---|
| Sample $\tau^*$ $mn$ times | 24 | 31 | 7 | 25 | 130 |
| Plan actions that minimize $C_{\tau^*}$ | 16 | 15 | 15 | 50 | 295 |
| Total for TIP Iteration | 40 | 46 | 22 | 75 | 425 |
| Evaluation for one episode | 5-20 | 2-10 | 2-5 | 3 - 18 | 100-500 |

Table 3: Runtime in seconds for the phases of the TIP algorithm on all problems when run on the authors' CPU machines. The ranges given show the runtime for the operation at the beginning and at the end of training, as some operations run longer as more data is added.

| Control Problem | Pendulum | Cartpole | $\beta$ Tracking | $\beta$ + Rotation | Reacher |
|---|---|---|---|---|---|
| Number of samples | 25 | 30 | 25 | 50 | 100 |
| Number of elites | 3 | 6 | 3 | 8 | 15 |
| Planning horizon | 20 | 15 | 5 | 5 | 15 |
| Number of iCEM iterations | 3 | 5 | 3 | 5 | 5 |
| Replanning Period | 6 | 1 | 2 | 1 | 1 |

Table 4: Hyperparameters used for optimization in MPC procedure for closed-loop control problems.

samples of the optimal trajectory) since the space of samples is 10-dimensional and uniform random sampling will not get good coverage of interesting regions of the state space.

### A.4 Cost Function Details

We set $n = 15$ and $m = 1$ for our Monte Carlo estimate of the cost function for each problem.

### A.5 Details on Planning Method

As mentioned in the main text, we use the iCEM method from Pinneri et al. [51] with one major modification: a fixed sample batch size. This is in order to take advantage of the JIT compilation features of JAX and avoid recompiling code for each new batch size.

In Tables 4 and 5, we present the hyperparameters used for the planning algorithm across each problem. The same hyperparameters were used for the TIP, MPC, $\text{EIG}_T$, DIP, sDIP, and sTIP methods. As recommended by the original paper, we use $\beta = 3$ for the scaling exponent of the power spectrum density of sampled noise for action sequences, $\gamma = 1.25$ for the exponential decay of population size, and $\xi = 0.3$ for the amount of caching.

## B    Description of Comparison Methods

We compare against 14 different methods across open and closed-loop problems. Of these, 7 used the same model and planning algorithm (including hyperparameters) as TIP and oTIP. **DIP** and **oDIP** use the cost function $C(\tau) = -\mathbb{H}\left[T(S') \mid D\right]$ and **sDIP** (summed DIP) uses the cost function $C(\tau) = -\sum_{i=0}^{h} \mathbb{H}\left[T(s_i, a_i) \mid D\right]$. These are all pure exploration methods, but DIP and oDIP are more sophisticated in that they plan for future observations with a large amount of *joint* information

| Control Problem | Nonlinear Gain 1 | Nonlinear Gain 2 | Lava Path |
|---|---|---|---|
| Number of samples | 50 | 50 | 25 |
| Number of elites | 6 | 6 | 4 |
| Planning horizon | 10 | 10 | 20 |
| Number of iCEM iterations | 6 | 8 | 6 |

Table 5: Hyperparameters used for optimization in MPC procedure for open-loop control problems.

as opposed to sTIP which sums the individual information expected at each timestep. oDIP is simply the open loop variant of DIP. **EIG**$_T$ uses the same objective as sDIP but operates in the TQRL setting, querying points that approximately maximize the predictive entropy of the dynamics model. **BARL** similarly operates in the TQRL setting but uses the $\text{EIG}_{\tau*}$ acquisition function from Mehta et al. [40]. We use the authors' implementation of that work for comparison. **MPC** uses $C_g$ from (4) and plans to directly maximize expected rewards. This method can be seen as quite similar to Kamthe and Deisenroth [34] and a close cousin of Deisenroth and Rasmussen [21] in that it optimizes the same objective with a similar model. **oMPC** is simply the open loop variant of MPC.

Besides these methods which directly compare cost functions, we include 8 additional baselines from published work. **PETS** is a method given in Chua et al. [15] which uses a similar cross-entropy based planner and a probabilistic ensemble of neural networks for an uncertainty-aware estimate of the dynamics. PETS also plans to minimize $C_g$. **HUCRL** [16] learns a policy via backpropagation through time using a hallucinated perturbation to the dynamics that maximizes discounted rewards subject to the one-step confidence interval of the dynamics. HUCRL also uses a probabilistic ensemble of neural networks. Using the same implementation we also tested Thompson Sampling (**TS**), which acts optimally according to a network drawn from the posterior over models, and **BPTT** which plans to minimize $C_g$ using a neural network policy and backpropagation through time. BPTT can also be viewed as a cousin of PILCO [21] as it attempts to take stochastic gradients of the expected cost. We also compare against **SAC** [27], **TD3** [23], and **PPO** [58]. SAC uses entropy bonuses to approximate Boltzmann exploration in an actor-critic framework. TD3 and PPO include various tricks for stable learning and add Ornstein-Uhlenbeck noise in order to explore.

For our FEEF implementation, we took hyperparameters from the most similar comparison environments in that paper and used them for our results. We tried several values for 'expl_weight' in order to se whether we were inadequately balancing exploration and exploitation. Ultimately we saw an 'expl_weight' of 0.1 was the best value.

We used the author's implementation of RHC. RHC makes strong assumptions on the form of the reward function by assuming that all problems are regulation problems where the goal is to drive the system to a given state and keep it there (with some cost for actuation). We were able to pass the targets for all of our problems (which may change between episodes) to the RHC controller. We did a light hyperparameter search tuning the number of random Fourier features used in the Bayesian linear model in this method. Ultimately we were disappointed in the performance of RHC when applied to our problems. We believe that this might be due to its undirected uncertainty sampling objective and relatively constrained model of environment dynamics.

## C   Description of Control Problems

### C.1   Plasma Control Problems

The plasma control problems are based on controlling a tokamak, a toroidally shaped device for confining a thermonuclear plasma using magnetic fields. Achieving net positive energy from fusion requires confining a plasma at high enough temperature and density long enough for hydrogen isotopes to collide and fuse. However, as the temperature and density are increased, a wide variety of instabilities can occur which degrade confinement, leading to a loss of energy. Full physics simulation of tokamak plasmas requires 10s-1000s of CPU hours to simulate a single trajectory, and often require hand tuning of different parameters to achieve accurate results. Following the work of Abbate et al. [2], each of our plasma control problems used neural networks trained on data as the ground truth dynamics models. We used the MDSPlus tool [62] to fetch historical discharges from the DIII-D tokamak in San Diego [22]. In total, we trained our models on 1,479 historical discharges. The data was pre-processed following the procedure outlined in Abbate et al. [2]. We describe how each environment was constructed in more detail below.

$\beta$ **Tracking**   In this environment the goal is to adjust the total injected power (PINJ) of the neutral beams so that the normalized plasma pressure, $\beta_N$ (defined as the ratio of thermal energy in the plasma to energy in the confining magnetic fields), reaches a target value of 2%. Reliably controlling plasmas to sustain high performance is a major goal of research efforts for fusion energy, so even this simple scenario is of interest. The ground-truth dynamics model takes in the current $\beta_N$ and PINJ, the $\beta_N$ and PINJ at some $\Delta t$ time in the past, and the PINJ at some $\Delta t$ time in the future (we assume

that we have complete control over the values of PINJ at all times). Given these inputs, the model was trained to output what $\beta_N$ will be $\Delta t$ time into the future. In total, the state space is 4D and the action space is 1D. For this environment, we set $\Delta t = 200ms$, and we specify the reward function to be the negative absolute difference between the next $\beta_N$ and the target $\beta_N = 2\%$.

$\beta$ **+ Rotation Tracking**   This environment is a more complicated version of the $\beta$ tracking environment in several ways. First of all, the controller now must simultaneously track both $\beta_N$ and the core toroidal rotation of the plasma. To do so, the controller is also allowed to set the total torque injected (TINJ) of the neutral beams (DIII-D has eight neutral beam injectors at different positions around the tokamak, so it is generally possible to control both total power and total torque independently). Controlling both of these quantities simultaneously is of interest since rotation shear often results in better confinement and less chance of instabilities in the plasma [8, 25]. In addition, we assume a multi-task setting where the requested targets for $\beta_N$ and rotation can be set every trajectory. Specifically, the $\beta_N$ target is drawn from $U(1.5\%, 2.5\%)$ and the rotation target is drawn from $U(25, 125)$ krad/s every trajectory. These targets are appended to the state space.

The learned, ground-truth dynamics model is also more sophisticated here. In addition to the inputs and outputs used by the $\beta$ tracking environment model, the inputs for this model also include rotation and TINJ at times $t$, $t - \Delta t$, and $t + \Delta t$ for TINJ only. This model receives additional information about the plasma (e.g. the shape of the plasma); however, we have assumed these inputs are fixed to reasonable values in order to avoid partial observability problems. In total, the state space of this problem is 10D (targets plus current and past observations for $\beta_N$, rotation, PINJ, and TINJ) and the action space is 2D (next PINJ and TINJ settings).

### C.2   Robotics Problems

**Pendulum**   The pendulum swing-up problem is the standard one found in the OpenAI gym [10]. The state space contains the angle of the pendulum and its first derivative and action space simply the scalar torque applied by the motor on the pendulum. The challenge in this problem is that the motor doesn't have enough torque to simply rotate the pendulum up from all positions and often requires a back-and-forth swing to achieve a vertically balanced position. The reward function here penalizes deviation from an upright pole and squared torque.

**Cartpole**   The cartpole swing-up problem has 4-dimensional state (position of the cart and its velocity, angle of the pole and its angular velocity) and a 1-dimensional action (horizontal force applied to the cart). Here, the difficulty lies in translating the horizontal motion of the cart into effective torque on the pole. The reward function is a negative sigmoid function penalizing the distance between the tip of the pole and a centered upright goal position.

**Reacher**   The reacher problem simulates a 2-DOF robot arm aiming to move the end effector to a randomly resampled target provided. The problem requires joint angles and velocities as well as an indication of the direction of the goal, giving an 8-dimensional state space along with the 2-dimensional control space.

## D   Additional Results

Due to space constraints in the main paper, we omitted results for the methods sDIP and BPTT. The are included alongside the rest in Table 6. They are outperformed across the board by TIP.

## E   Additional Related Work

### E.1   Bayesian Exploration Techniques

Given unlimited computation and an accurate prior, solving the Bayes-adaptive MDP [53] gives an optimal tradeoff between exploration and exploitation by explicitly accounting for the updated beliefs that would result from future observations and planning to find actions that result in high rewards as quickly as can be managed given the current posterior. However, this is computationally expensive even in small finite MDPs and totally intractable in continuous settings. Kolter and Ng

| Environment | TIP | sTIP | DIP | sDIP | MPC | PETS | SAC | TD3 | PPO | FEEF | HUCRL | TS | BPTT | BARL | EIG$_T$ |
|---|---|---|---|---|---|---|---|---|---|---|---|---|---|---|---|
| Pendulum | **21** | 36 | 36 | 46 | 46 | 5.6k | 7k | 26k | 14k | 800 | >50k | >50k | >50k | **21** | 56 |
| Cartpole | 131 | 141 | 161 | 141 | 201 | 1.63k | 32k | 18k | >1M | >2.5k | >6k | >6k | >6k | **111** | 121 |
| $\beta$ Tracking | **46** | 76 | 276 | 131 | 76 | 330 | 12k | 17k | 39k | 300 | 480 | 420 | 450 | 186 | >1k |
| $\beta$ + Rotation | **201** | >500 | >500 | >500 | >500 | 400 | 30k | >50k | >50k | >2k | >5k | >5k | >5k | >500 | >1k |
| Reacher | **251** | >400 | >1k | >1k | 751 | 700 | 23k | 13k | >100k | >5k | 6.6k | 4.5k | 3.7k | **251** | >1.5k |

Table 6: **Sample Complexity Comparison of All Methods:** Median number of samples across 5 seeds required to reach 'solved' performance, averaged across 5 trials. We determine 'solved' performance by running an MPC policy (similar to the one used for evaluation) on the ground truth dynamics to predict actions. We record $> n$ when the median run is unable to solve the problem by the end of training after collecting $n$ datapoints. The methods in the rightmost section operate in the TQRL setting and therefore have more flexible access to the MDP dynamics for data collection.

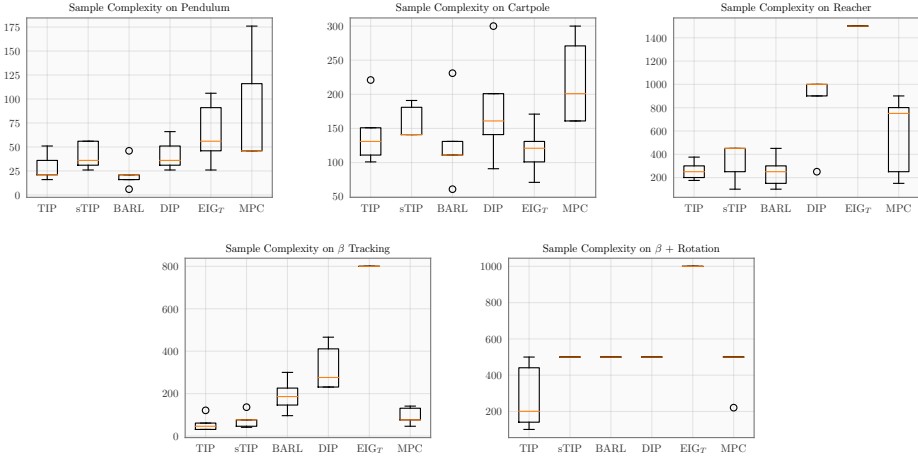

Figure 4: Box plots showing sample complexity figures across the 5 random seeds run. Each of these show for a given training run how many samples were needed to achieve the performance of an MPC controller given ground truth dynamics averaged across test episodes. We imputed the maximum number of samples for agents that failed to ever solve the problem on a given run.

[35] and Guez et al. [26] show that even approximating these techniques can result in substantial theoretical reductions in sample complexity compared to frequentist PAC-MDP bounds as in Kakade [33]. Another line of work [18, 19] uses the myopic value of perfect information as a heuristic for similar Bayesian exploration in the tabular MDP setting. Further techniques for exploration include knowledge gradient policies [57, 56], which approximate the value function of the Bayes-adaptive MDP and information-directed sampling (IDS) [54], which takes actions based on minimizing the ratio between squared regret and information gain over dynamics. This was extended to continuous-state finite-action settings using neural networks in Nikolov et al. [44]. Another very relevant recent paper [7] gives an acquisition strategy in policy space that iteratively trains a data-collection policy in the model that trades off exploration against exploitation using methods from active learning. Achterhold and Stueckler [3] use techniques from BOED to efficiently calibrate a Neural Process representation of a distribution of dynamics to a particular instance, but this calibration doesn't include information about the task. A tutorial on Bayesian RL methods can be found in Ghavamzadeh et al. [24] for further reference.

## E.2 Gaussian Processes (GPs) in Reinforcement Learning

There has been substantial prior work using GPs [52] in reinforcement learning. Most well-known is PILCO [21], which computes approximate analytic gradients of policy parameters through the GP dynamics model while accounting for uncertainty. The original work is able to propagate the first 2 moments of the occupancy distribution through time using the GP dynamics and backpropagate gradients of the rewards to policy parameters. In [69], a method is developed for efficiently sampling functions from a GP posterior with high accuracy. One application show in their work is a method of using these samples to backpropagate gradients of rewards through time to policy paramters,

which can be interpreted as a different sort of PILCO implementation. Most related to our eventual MPC-based method is [34], which gives a principled probabilistic model-predictive control algorithm for GPs. We combine ideas from this paper, PETS [15], and the ability to sample posterior functions discussed above to give our eventual MPC component as discussed in Section 4.1.