# OpenReview forum: "Exploration via Planning for Information about the Optimal Trajectory"
_NeurIPS.cc/2022/Conference — NeurIPS 2022 Accept_

### Official Review · Reviewer_iWun · 2022-07-05

**Rating:** 5
**Confidence:** 3
**Soundness:** 2 fair
**Presentation:** 1 poor
**Contribution:** 3 good

**Summary:**

The authors develop Bayesian Model-Predictive Control in which a new cost-function is proposed. The idea of the cost-function goes back to the information gain notion. The aim of the paper is to make the algorithm more sample efficient.

**Questions:**

1. There is no description what is $P(T|D)$ in formula (2). Also it is not clear how distributions differ for open-loop and closed-loop problems.

2. In the end of Algorithm 1, it is unclear how to create policy $\pi_g$ for posteriory distribution $P (T' | D)$.

3. On lines 203 and 204, the use of the notation $S$ for pairs, and $S'$ for one state does not seem quite intuitive to understand. Maybe it's better to use another letter instead of $S$?

4. In formula (5), it is not clear what is the result of $D \cup S'$  and what symbol $p(S'|S,D)$ stands for. When $D$ – is the set of triples, $S$ – is the set of pairs, $S'$ - is the set of single states. What is the connection between $D$ and $S$?

5. In formula (6), it is unclear what is the result of union $D \cup \tau^*$, when $D$ – is the set of triples, $\tau^*$ is the set of pairs of states and actions and a final state.

6. Perhaps, since line 207 gives an idea of a new cost function, it would be better to give a simple example when it works better than the previous ones.

7. In line 210 the non-obvious equality $C_{τ^*}(S) = −I(S', τ^*) = −I(τ^*, S')$ is given. It would be better to prove it or provide the link to the proof.

8. In formula (7) it is unclear what is the result of union $D \cup \tau^∗_{ij}$, when $D$ – is the set of triples, $\tau^∗_{ij}$ - is the set of pairs of states and actions and a final state.

9. In algorithm 2, it is unclear how to compute the joint posterior covariance matrices. If a gaussian process is used for this, then it would be better to specify what is the kernel function.

10. What hyperparameters were used to run experiments and baseline algorithms (so that they could be repeated)?

**Limitations:**

The article does not have potentially negative societal impact.

**Strengths And Weaknesses:**

The problem considered in the article seems relevant and the algorithms proposed by the authers seems interesting and original. The experiments show that, apparently, the algorithms are the most effective in terms of using samples. Nevertheless, in my opinion, the Weakness of the article is its descriptive part. To describe the ideas, the authors use mathematical language very carelessly, which can cause the reader serious problems in understanding the article (for more information, see the questions section).  Thus, the article seems to be promising, but requires significant refinement, in particular, in the mathematical presentation – it should be more rigorous.

---

> ### Author Response · Authors · 2022-08-02
> **Response to Reviewer iWun**
>
> Thank you for your review. We’re glad you found our method interesting and original and agree that the TIP algorithm is the most effective on our intended goal of sample efficiency. We’ll address your questions in order below, starting with a few edits we’ve made to our mathematical presentation.
>
> ### **Questions**
>
> 1. Thanks for pointing out that we didn’t explicitly define $P(T|D)$. As is typically the case when GPs are used, $P(T|D)$ is the posterior distribution over the transition function $T$, given the GP prior and data observed. We’ve added a note to our preliminaries explicitly stating this.
>
> 2. We take $\\pi_g$ to be the MPC policy (i.e. the policy produced by running the MPC algorithm) given in Equation 3, with the cost function $C_g$ (described in line 233). We have made an edit  explicitly defining this around line 170.
>
> 3. We changed $S$ to $X$ and $\\tilde{\\mathcal{S}}$ to $\\mathcal{X}$ for the domain of our cost function throughout our paper. Thanks for the suggestion!
>
> 4. (+ 5., 8.) The confusion here is that variables $S’$ and $\\tau^*_{ij}$ are themselves made of dynamics data in the sense that they encode transitions that come from our dynamics.
>
> In this work we assume the unions between the dataset D and other mathematical objects such as the trajectories $\\tau_{ij}^*$ or the next states $S’$ given $X = (s_i, a_i)$ and some data do the natural thing and construct the state, action, and next state triples that allow them to fit as elements of a set of data. We added a sentence at the end of the preliminaries clarifying the meaning of this operation.
>
> 6. One classic example where naive exploration cost functions such as $C_e$ work poorly is the noisy-tv problem, where the agent is presented with unpredictable noise that is irrelevant to the task. TIP will correctly identify that this information does not affect the dynamics which matter to the task and ignore it.
>
> 7. This equality is by the conditional entropy definition of the mutual information I and the fact that it is symmetric.
>
> However, we felt that a derivation is unnecessary here since these are both well-known properties of mutual information (for example, listed as properties on the [wikipedia page](https://en.wikipedia.org/wiki/Mutual_information#Properties) on mutual information).
>
>
>
> 9. In line 220 we say explicitly that this covariance is easy to calculate using a GP and give a reference. We also mention in Section A.3 that we use a squared exponential kernel and give details on the kernel fitting procedure.
>
> 10.  We give full hyperparameters for TIP in Tables 4 and 5 in the appendix. As we have over a dozen comparison methods we believed it would be excessive to include tables of hyperparameters for each of the other methods used. We will release code for our method and all comparison methods which will allow for the experiments to be easily repeated by anyone interested.
>
> Thanks again for your feedback on the paper. Comments like these allow us to make concrete improvements that help people understand our work more clearly. We hope that our modifications and clarifications have resolved your questions.
>
> Finally, if you find our response satisfactory, we respectfully ask that you consider increasing your score. If it is still unsatisfactory, please let us know if there is anything else that we can do or clarify to improve this paper.

---

### Official Review · Reviewer_ZhTc · 2022-07-10

**Rating:** 7
**Confidence:** 4
**Soundness:** 4 excellent
**Presentation:** 3 good
**Contribution:** 3 good

**Summary:**

The paper aims at improving data efficiency for RL, targeting real systems where data acquisition is normally costly.
The authors consider model-based control for MDPs with explicitly modelled dynamics.
Control is relegated to MPC, the procurement of parametric (amortized) policies is left out of scope.
The focus is on obtaining useful data for learning the dynamics model as fast as possible, so that MPC can successfully control the agent.
The method shows significant improvement in data efficiency in 5 different environments with simple to moderately complex dynamics.

The main premise is that explorative actions are selected such that maximum information (in the sense of infogain, mutual information, MI) is gained for the optimal future trajectory $\tau^*$ under the **current** dynamics model, noting that this is better than plain max. entropy RL, as it takes the task into account.
This is captured by the *Expected Information Gain* (EIG) objective, which has been proposed for the RL setting in prior work [1, 2].
In this paper, the objective is extended to account for the acquisition of data in sequence, i.e. in the form of rollouts in the real system.
To me it appears that this is a step forward from the cited prior work [1], where the system could be queried arbitrarily (at any state of the agent, coined the TQRL setting), which is unrealistic on real hardware.
One implication of this treatment is that mutual information is evaluated between the optimal future trajectory $\tau^*$ and joint sets of future states corresponding to candidate future control sequences, i.e. it serves as the objective for evaluating MPC rollouts and accounts for possible information overlaps into the future.
The method is called *Trajectory Information Planning* (TIP).

**References**

[1] Mehta, V., Paria, B., Schneider, J., Ermon, S. and Neiswanger, W., 2021. An Experimental Design Perspective on Model-Based Reinforcement Learning. arXiv preprint arXiv:2112.05244.

[2] Neiswanger, W., Wang, K.A. and Ermon, S., 2021, July. Bayesian algorithm execution: Estimating computable properties of black-box functions using mutual information. In International Conference on Machine Learning (pp. 8005-8015). PMLR.

**Questions:**

- The **MPC** ablation variant from section 5, i.e. a GP + MPC with no exploration considerations, seems to perform very well in terms of data efficiency. This is expected, as the interpolation properties of GPs are a strong prior and aid generalization when the system dynamics behave nicely in the Lipschitz sense (PILCO is a good example of this). Do you believe the TIP objective would work just as well if this aspect was not there, i.e. without the inherently fast adaptation of a GP due to its interpolation properties?
- Can you further elaborate on why TIP performs better than BARL for the plasma control tasks? The text mentions that this could be due to the inefficiency of sampling a finite candidate set for BARL, do you believe that ends up being worse than the inherent proximity of sequential trajectory states acquired by TIP?
- I don't think I agree with the statement that the median amount of data used by algorithms to solve the task is equivalent to an MPC controller with ground-truth dynamics, as stated in sec. 5 -- what is the justification?
- Have you considered using a critic, to account for infinite horizons?

Also see my remarks above.


**Limitations:**

I don't see any major negative societal impact. The authors acknowledge the main methodological limitations of the method in the conclusion, although these could be emphasized further.

**Strengths And Weaknesses:**

**Strengths**
- The posed exploration objective appears sound and widely applicable. I also find it fitting, as it is about reducing the predictive entropy of the optimal trajectories MPC will actually produce, by virtue of training the dynamics model on data that is informative about them.
- While the core idea of the objective is not novel (see [1]), I believe the extension to realistic data acquisition, i.e. in sequences, is a step forward.
- The authors make it abundantly clear that information overlap in the acquired data sequences needs to be accounted for, and the method complies with that.
- Motivation is easy to follow across the manuscript, exposition is clear.

**Weaknesses**
- The method hinges on the assumption that the posterior over the dynamics model can be updated in closed form (eq. 6 and 7) and that the entropy of predictions is easy to evaluate. This is required for the evaluation of the posed MI objective. Therefore the method limits itself to GP dynamics (just like in [1]), black-box neural dynamics models would encumber the evaluation of the objective.
- GPs can be limiting both in terms of computation and expressivity for high-dimensional state spaces;
- The authors use a variation of the cross-entropy method (CEM) for MPC optimization. MC-based optimization techniques are easy to apply, but might scale poorly to high-dimensional control spaces.
- While I find the selection of experimental tasks reasonable, a lot of the environments are low-dimensional and with nice and smooth dynamics. How would the method fair on something more complex, like the ant locomotion task? Would the GP still be able to capture the transition accurately? For me it remains an open question whether the scheme is applicable to more complex systems (i.e. scalability).
- The mutual information objective is formulated for the optimal trajectory $\tau^*$ w.r.t. the current dynamics posterior. I think it is important to highlight that the "optimal trajectory" in that sense is not the ideal trajectory, but the best one available based on the current model fit. So overall acquisition success is still subject to the reachability properties of the state-space of the system.

Overall I found this is a good contribution that explores an interesting idea already introduced in prior work [1] further. Thus I lean towards acceptance.

**Minor remarks**
- The qualifier "simply" appears too often in the text (11 times).

---

> ### Author Response · Authors · 2022-08-02
> **Response to Reviewer ZhTc**
>
> Thank you for your detailed and helpful review. We appreciate the positive feedback and hope to address your concerns in a way that improves the quality of our submission. We largely agree with your summary of the paper and its contributions.
>
> Please see our note to all reviewers regarding high dimensional RL problems as it was a concern in common.
>
> ### **Regarding** $\tau^*$ :
> Under our assumptions $\tau^*_{ij}$ is the optimal trajectory for a particular sample from the posterior $T_j \sim P(T | D)$ and start state $s_i$. As long as the true dynamics are in the support of the posterior then the support of the true optimal trajectory (the states visited by executing an optimal policy from the start state distribution on the GT dynamics, which are by definition reachable) should be a subset of the support of $P(\\tau^*|D)$ since the GT dynamics $T$ must be in the support of $P(T|D)$. The core objective of the cost function is to identify the dynamics needed to plan the optimal trajectory as quickly as possible.
>
> That being said, we are certainly constrained to only collect data from places that are reachable by the exploration policy. This is an inherent constraint of the rollout RL setting and we do not avoid this.
>
> ### **How much of TIP's performance is attributable to GPs?**
> GPs have been shown in PILCO[1] and many other works[2][3] to be notably sample efficient dynamics models in regimes where they are statistically and computationally capable of fitting the MDP dynamics. As mentioned elsewhere they also allow us to compute an estimate of $C_{\\tau^*}$ using only Monte Carlo and an assumption of a good planner on known dynamics and we aim to extend to more scalable models using other techniques to compute a similar quantity.
>
> PETS [4] uses probabilistic neural ensembles and a stochastic MPC algorithm and is comparable to PILCO[1] and our MPC baseline,  similar methods with a GP model--in our experience the GP has better sample complexity on the more limited set of problems with well-shaped reward functions and dynamics that are amenable to being fit by a GP but fails in many problems where PETS succeeds due to a more flexible model. We believe that an approximation of the $C_\\tau^*$ cost function that works with neural network models would explore more effectively than simply exploiting the model as in PETS or our MPC baseline; however we leave such extensions for future works.
>
> ### **Why does TIP sometimes outperform BARL?**
> BARL optimizes its acquisition function by simply uniformly sampling the domain, evaluating the acquisition function at each sampled point, and choosing the maximum. This will work for very low-dimensional problems, but as the dimensionality increases to the 10D beta + rotation problem, this very simple optimization algorithm becomes much less effective. The combination of an iterative optimization algorithm (CEM) and the simple fact that TIP is forced to start from the initial state help mitigate the difficulty of optimizing the (nonconvex) mutual information quantity.
>
> ### **Odds and ends**
> I think our sentence in Section 5 is unclear. We define solving the problem as reaching performance equivalent to an MPC controller given GT dynamics. This is a definition used in our evaluation procedure and not a claim of anything further. We have updated the wording to make this less ambiguous.
>
> We are actively looking into various approaches for including a terminal value function for the cost function we introduce in the paper. I believe this could both improve the scalability of the method (as NN models would work) and its ability to handle infinite horizon. Thanks for the suggestion!
>
> We appreciate your review and feedback!
>
> [1] PILCO: A Model-Based and Data-Efficient Approach to Policy Search, Deisenroth & Rasmussen, ICML 2011
>
> [2] Data-Efficient Reinforcement Learning with Probabilistic Model Predictive Control, Kamthe and Deisenroth, AISTATS 2018
>
> [3] Gaussian Processes in Reinforcement Learning, Malte & Rasmussen, Neurips 2003
>
> [4] Deep Reinforcement Learning in a Handful of Trials using Probabilistic Dynamics Models, Chua et al, Neurips 2018

---

### Official Review · Reviewer_QRq7 · 2022-07-11

**Rating:** 6
**Confidence:** 5
**Soundness:** 2 fair
**Presentation:** 3 good
**Contribution:** 2 fair

**Summary:**

The paper extends Bayesian Active Reinforcement Learning (BARL) [38] by lifting its Transition Query RL (TQRL) assumption (TQRL means that the agent can sample individual transitions $(s,a,r,s')$ by querying the simulator at an arbitrary $(s,a)$ pair). Instead, only state-action pairs from collected trajectories are used in this paper.

The main contribution is Algorithm 2 for computing the cost function $C_{\tau^*}(S)$, the negative joint expected information gain about the optimal trajectory. The average cost is subsequently optimized via the cross-entropy method (CEM). The computation complexity of one iteration is derived, and experiments on 5 environemnts for closed-loop policies (the algorithm applied in MPC fashion) and 3 environments for open-loop policies are reported.

The proposed method Trajectory Information Planning (TIP) is shown to perform comparably or even better than BARL in the sense of requiring fewer samples. The fact that TIP is even better than BARL is surprising since BARL can sample the state-action space more freely, but the authors hypothesize that this is because BARL samples uniformly whereas TIP has a more sophisticated scheme and forces initialization at the start distribution.

To demonstrate that explicit exploration is useful, TIP is shown to require fewer samples than the methods which don't explicitly encourage exploration (MPC, PETS, SAC, TD3, PPO).

Furthermore, TIP is compared to a few baselines — HUCRL and TS — that rely on upper confidence bound (UCB) and Thomson sampling (TS) over the dynamics and shown to provide better exploration by requiring significantly fewer samples.

Finally, a few ablations of the proposed method are considered that either only use the cost function $C_g$ (used in MPC) or the entropy objective $C_e$ (used in DIP, EIG$_T$). A variant of TIP where the information gain is split over time steps called sTIP is also considered. All these methods perform comparably, according to Table 1, however TIP shows the best sample complexity.

**Questions:**

1) In Fig. 3, "Model MSE on Random Set" is bigger than "Model MSE on Current MPC" by 4 orders of magnitude. This seems to imply that the model is quite bad outside of the observed data. How should one interpret it? Can one then rely on such model?
2) Table 1 shows only the median value over 5 runs. It would be usefull to see the whole distributions, at least in the appendix. E.g., display a boxplot for each environment. Otherwise, it is hard to judge how reliable the numbers in Table 1 are and what is the variance in these results.
3) MPC seems to perform quite comparably to TIP and it requires significanlty fewer samples than SAC/TD3/PPO/HUCRL/TS, despite the fact that it has no notion of exploration and is only optimizing the cost function $C_g$. This seems to indicate that explicit exploration may not be so necessary in the considered tasks or that MPC somehow performs sufficient exploration anyways. It would be worth providing a discussion of this result.
4) DIP appears very similar to MPC in Table 1 despite the fact that they are using completely different objective functions. DIP is only using the entropy objective $C_e$ whereas MPC is only maximizing the reward  $C_g$. Such similarity seems to indicate that the reward is somehow correlated with the exploration in the considered tasks. Or is there a different explanation?
5) Results for BARL are better in the BARL paper (Table 1 here https://arxiv.org/pdf/2112.05244.pdf) compared to the results in Table 1 in the present paper. What is the reason for that?


**Limitations:**

Technical limitations are described sufficiently. There is no discussion of the negative societal impact. The authors could indicate potential dangers of active exploration in applications of RL.

**Strengths And Weaknesses:**

Strengths
- clarity: well-written paper, with good structure and clear presentation

Weaknesses
1) Significance: medium to low. Too dramatic and misleading claims in the abstract
    - "2-200x fewer samples compared to 14 baselines" — the 200x improvemnt is w.r.t. methods that don't compute information gain (PETS/SAC/TD3/PPO). Such comparison does not seem fair. There has been a lot of algorithms that propose various version of information gain and artifical curiosity, therefore a comparison should be with respect to those algorithms or at least w.r.t. BARL or MPC with entropy objective. Compared to these algorithms, the contribution is still valuable but the statement would be more sober.
2) Originality: fair
    - Reading the paper briefly makes an impression that the whole approach of minimizing the information gain about the optimal trajectory is new. In particular, this is also claimed in the abstract. However, all the derivations and ideas are already presented in BARL [1]. The main contribution of this paper compared to BARL is to compute the InfGain using only sampled trajectories. It should be made much more clear what is the contribution of the present paper and what is actually novel here.
3) Quality: fair. A lot of evaluations but comparison to wrong baselines
    - As mentioned above, the baselines are either from a different category (SAC/TD3/PPO) or don't use InfGain (MPC/PETS) or seem to be badly tuned or not applicable (HUCRL/TS) or are version of the proposed algorithm (sTIP, DIP, EIG$_T$). On the other hand, a variety of exploration bonuses and surrogates for the InfGain have been proposed, e.g., [2,3]. In particular, the proposed method appears very similar to [4] and [5] and therefore either a comparison would be desirable or a statement why such comparison is not necessary.


References \
[1] Mehta, V., Paria, B., Schneider, J., Ermon, S., & Neiswanger, W. (2021, September). An Experimental Design Perspective on Model-Based Reinforcement Learning. In International Conference on Learning Representations. \
[2] Shyam, P., Jaśkowski, W., & Gomez, F. (2019, May). Model-based active exploration. In International conference on machine learning (pp. 5779-5788). PMLR. \
[3] Pathak, D., Gandhi, D., & Gupta, A. (2019, May). Self-supervised exploration via disagreement. In International conference on machine learning (pp. 5062-5071). PMLR. \
[4] Schultheis, M., Belousov, B., Abdulsamad, H., & Peters, J. (2020, May). Receding horizon curiosity. In Conference on robot learning (pp. 1278-1288). PMLR. \
[5] Tschantz, A., Millidge, B., Seth, A. K., & Buckley, C. L. (2020). Reinforcement learning through active inference. arXiv preprint arXiv:2002.12636.

---

> ### Author Response · Authors · 2022-08-02
> **Response to Reviewer QRq7**
>
> Thank you for your detailed review. We appreciate your feedback and aim to satisfy your concerns while also improving the paper. We broadly agree with your summary of the paper’s contents. We’ll address your concerns below and highlight the changes we made in response to them, which are now present in the updated version of the paper.
>
> ### **Claims about algorithm performance**
> We understand that the claims in the abstract and introduction are imprecise and might give the wrong impression and have changed them to be more specific about how much we improve against different types of algorithm so as to be more clear about the improvements.
>
> ### **Originality**
> Though the idea of maximizing information gain with respect to the optimal trajectory is introduced in the BARL paper, we believe that the generalization of the acquisition function from a single point to a joint information gain over a batch used for planning is one which wasn’t obvious in the BARL paper. We see similar contributions of this type in multiple other works in the BO literature [1][2]. We address the scope of our novelty on line 183 saying:
>
> > “this is the same overall goal as that of Mehta at al. , where the $EIG_{\\tau^*}$ acquisition function was introduced… However, in this paper we generalize this acquisition function in order to allow for sequential information collection that accounts for the redundant information that could be collected between timesteps.”
>
> However, we also added an additional citation to BARL to the introduction and some text in the related work emphasizing the connection.
>
> ### **Baselines**
> The exploration algorithms you cited [3][4][5][6] can be broken down into 2 categories. We see [3][4] and [5] as pure exploration algorithms that don’t consider the task being solved and are closely related to our DIP baseline, though they use various tricks to approximate information gain with neural networks or in the case of [5] a Bayesian linear model. As we mention on line 168, [3] is essentially an algorithm that optimizes an EIG criterion (equation 4 from that paper) that can be shown to be equivalent to $C_e$ in our paper. [4] is similarly optimizing the model disagreement in a predictive ensemble, which is an analogous quantity to the posterior predictive variance of a GP model and [5] is quite similar, using a curiosity objective over future timesteps (equation 2 in that paper is very close to the DIP objective). We believe that DIP is the best member of this family of algorithms to compare against as it removes the modeling aspects (especially since GPs perform well) and allows the two exploration strategies to be compared with the same dynamics estimation and planning algorithm. We added a short note to our updated draft clarifying this.
>
> [6] is a paper that we hadn’t seen before, and we agree that it is of a similar spirit. We have conducted additional experiments where we ran the FEEF algorithm from [6] on our benchmark tasks under several hyperparameters similar to those used in their experiments. TIP solves the problem with a small fraction of data that FEEF uses on each task and FEEF fails to solve 3 of our 5 tasks. We have included these results in our paper.
>
> We believe that HUCRL and TS are similar to TIP and FEEF as methods which attempt to gather task-relevant information about the dynamics and are therefore specifically relevant comparisons. We agree that the results we observe from these algorithms are disappointing. However, we spent significant effort attempting to tune the authors’ implementations of HUCRL/TS, and even consulted with the first author of the HUCRL paper [7] over Zoom to ask for advice on tuning hyperparameters. After this effort we decided to include the best results we could achieve using the code provided.
>
> Finally, in order to make FEEF fit and with the aim of deemphasizing the comparison to the model-free methods, we moved PPO and TD3 to the appendix.

---

> > ### Author Response · Authors · 2022-08-02
> > **Continued**
> >
> > ### **Questions:**
> > Q1: The correct interpretation of the modeling error is that the error on the points queried by the planner on the ground truth dynamics is much smaller than the error on the randomly chosen points in the domain. As the domain could be arbitrarily larger than the support of the optimal trajectory, it is likely that predictions would be poor on a uniform sample of the much larger set. We believe that this is an unavoidable consequence of task-oriented model-based reinforcement learning with very limited data and not unique to our method.
> >
> > Q2: Though we include error bars in our plots in Figure 3, we agree that box plots are a good way to present the statistical information around performance. We have added these plots to the appendix in the updated version of our paper. Thanks for the suggestion!
> >
> > Q3: We disagree with the premise that MPC performs comparably. MPC takes 2.2x, 1.7x, 1.5x, >2.5x, and 3x more samples than TIP to solve our benchmark problems. However, model-based reward maximization algorithms using GPs have consistently shown strong results with few samples on simple to moderately complex tasks. We are therefore not too surprised that MPC performs well on the examples here, especially since we made sure the planner was sufficiently strong to find a good solution to each problem.
> >
> > Q4:  The problems that DIP performs well on are the lower dimensional problems where it is possible to simply fully explore the state space and solve the problem at test time. Once the dimensionality increases it is no longer possible to easily fully explore the state space and therefore DIP doesn’t perform well. MPC is able to be more directed, but suffers due to focus on maximizing reward rather than directly optimizing for additional information. As a result,  it doesn't explore as quickly as TIP but, at the same time, doesn't get stuck like DIP.
> >
> > Q5: In the BARL paper, BARL performs slightly better on Pendulum (16 datapoints vs 21) and cartpole (91 points vs 111), exactly the same on Reacher, and substantially better on the beta tracking environment (96 vs 186), in comparison with the one we used in this paper. We believe the pendulum and cartpole results are due to small fluctuations in performance and the fact that we evaluate performance periodically as data is acquired. We are sure that for the classic control environments we used the same ones as in the BARL paper.
> >
> > For the beta tracking environment, we trained the dynamics using our plasma data to try to be similar to those from the BARL paper but there ended up being differences in the ultimate environment. We added a comment to our description of the control problem clarifying this).
> >
> > Finally, we hope our response and updates to the paper addressed most of your concerns with the submission. If so, we respectfully request that you increase your score. Please let us know if there is anything else we could clarify. Thanks again for your feedback as it was instrumental in improving this paper.
> >
> > [1] Batch Bayesian Optimization via Local Penalization, Gonzalez et al, AISTATS 2016
> >
> > [2] The Parallel Knowledge Gradient Method for Batch Bayesian Optimization, Wu & Frazier, Neurips 2016
> >
> > [3] Shyam, P., Jaśkowski, W., & Gomez, F. (2019, May). Model-based active exploration. In International conference on machine learning (pp. 5779-5788). PMLR.
> >
> > [4] Pathak, D., Gandhi, D., & Gupta, A. (2019, May). Self-supervised exploration via disagreement. In International conference on machine learning (pp. 5062-5071). PMLR.
> >
> > [5] Schultheis, M., Belousov, B., Abdulsamad, H., & Peters, J. (2020, May). Receding horizon curiosity. In Conference on robot learning (pp. 1278-1288). PMLR.
> >
> > [6] Tschantz, A., Millidge, B., Seth, A. K., & Buckley, C. L. (2020). Reinforcement learning through active inference. arXiv preprint arXiv:2002.12636.
> >
> > [7] Efficient Model-Based Reinforcement Learning through Optimistic Policy Search and Planning, Curi et al, Neurips 2020
> >
> > [8] PILCO: A Model-Based and Data-Efficient Approach to Policy Search, Deisenroth & Rasmussen, ICML 2011

---

> > > ### Author Response · Authors · 2022-08-09
> > > **Update: Additional Baseline**
> > >
> > > We have additionally implemented and updated our paper with results from comparing another suggested baseline method, RHC [1], on our benchmark environments. Though we did a moderate amount of hyperparameter tuning, RHC doesn’t perform well on our benchmark environments, either failing to solve the problem or solving the problem using many times more data than TIP. We hope that including both methods you noted as most relevant to our work in our updated version of the paper has addressed your concerns as to its quality.
> > >
> > > [1] Schultheis, M., Belousov, B., Abdulsamad, H., & Peters, J. (2020, May). Receding horizon curiosity. In Conference on robot learning (pp. 1278-1288). PMLR.

---

> > ### Comment · Reviewer_QRq7 · 2022-08-09
> > **Score Raised**
> >
> > I thank the authors for addressing my questions and comments. The updates made to the manuscript have resolved my concerns. Therefore, I raised my score.

---

### Official Review · Reviewer_4S7A · 2022-07-19

**Rating:** 6
**Confidence:** 4
**Soundness:** 2 fair
**Presentation:** 3 good
**Contribution:** 2 fair

**Summary:**

This paper presents a method for exploration in MBRL techniques by proposing a method called trajectory information planning. They present an algorithm to maximize the expected information gain for the trajectory during planning. The method have been demonstrated on several control problems and compared with some baselines.

**Questions:**

1. How would your method work for high dimensional tasks? How would you method compare against a Stochastic MPC approach? like chance constrained MPC? or Tube MPC kind of approaches. In general, they tend to work well on classical control systems.
2. How would your method scale to high-dimensional systems? Did you try implementing your method on acrobot or humanoid kind of systems?

**Ethics Review Area:**

["I don’t know"]

**Limitations:**

1. I think authors can try and address some limitations with respect to computations and scaling of the method to high dimensions. That could be useful insight for the readers.

**Strengths And Weaknesses:**

Strengths:
1. The information gain metric is interesting and seems novel to me.
2. The authors can show improvement over some chosen baseline methods
Weaknesses:
1. I think the evaluation of the paper needs more work.
2. Would it be possible for authors to perform comparison on some high-dimensional problems? Also, I think it is unfair to compare to model-free methods as they are known to slow compared to model-based method on these classical control tasks. If you would like to compare your method to model-free methods, please select some tasks where the SOTA MBRL techniques dont perform good compared to model-free methods, and try to show that your method succeeds. Otherwise the comparison seems unfair.

---

> ### Author Response · Authors · 2022-08-02
> **Response To Reviewer 4S7A**
>
> Thank you for your review of our paper. We appreciate the positive feedback on our method and results.
>
> Please see our note to all reviewers regarding high dimensional RL problems as it was a concern in common.
>
> We’ll address your other comments below:
>
> ### **Model-Free Methods**
> We agree that it is widely accepted that model-free methods have comparatively poor sample complexity on classic control problems where dynamics are smooth. However, we also evaluate all methods on plasma control environments, which are typically solved by model-free methods[1] [2] for a number of reasons. We believe that in general it is informative to include results that could plausibly be used to solve many RL problems even when they are not necessarily the strongest in the dimension being analyzed. Given that we include seven model-based closed-loop baselines and three model-free ones, we believe that our set of comparative methods is robust. However, in response to another reviewer’s concern, we added a related directed exploration technique, FEEF, to our experiments and took the opportunity to deemphasize the comparison to model-free methods by moving PPO and TD3 to the appendix.
>
> [1] Jaemin Seo et al 2021 Nucl. Fusion 61 106010
>
> [2] Magnetic control of tokamak plasmas through deep reinforcement learning, Degrave et al, Nature
>
> ### **Robust MPC Methods**
> Both chance-constrained MPC and tube MPC are robust MPC methods which take the model uncertainty into account when synthesizing trajectories for execution and can achieve various guarantees for performance. These methods are in some sense complementary to the progress in TIP. TIP is a method of exploring the dynamics to solve a problem using few samples. One configuration that could make sense is to use TIP to obtain data during an agent’s exploration phase and then a robust MPC method during deployment to achieve good performance given the remaining model uncertainty. Further integrating these methods by exploring explicitly to find points that improve the robust performance of the MPC controller is an interesting direction we leave for future work.
>
> Finally, if our response satisfied your concerns we respectfully ask that you consider increasing your score. Please let us know if there is anything else that we can do or clarify to improve this paper. Thanks again for helping us improve the paper.

---

> ### Comment · Reviewer_4S7A · 2022-08-09
> **Post rebuttal comment**
>
> Dear Authors -- thanks very much for answering the questions. In my opinion, I think the evaluation of your method needs more work. You need to show performance on more complex environments. Inverted pendulum and reacher is too simplistic, even if you show more underactuated domains like acrobot or cart-pole would work.

---

> > ### Author Response · Authors · 2022-08-09
> > **Response to Post Rebuttal Comment - Cartpole Results**
> >
> > Thank you for your reply! Please note that one of the two suggested environments that you listed in your reply, cart-pole, is indeed in our experiments (see row 2 of Table 1).
> >
> > We agree that this environment is more challenging (sparse rewards, underactuated dynamics, etc.), and indeed, several of the baselines we compared our method to could not solve this task (HUCRL, TS, FEEF, RHC).

---

> > > ### Comment · Reviewer_4S7A · 2022-08-09
> > > **update**
> > >
> > > Thanks for pointing that out. I am suggesting that you perform a thorough evaluation of your method which would make your paper more useful for readers. I am updating my score to weak accept.

---

### Author Response · Authors · 2022-08-02
**High-Dimensional Environments**

Many of the reviewers asked for additional evaluations of our method on high-dimensional environments. We agree that they are an interesting and important subset of RL tasks and that scaling to high-dimensional environments up to text and images is an important goal of the field.

Our work focuses on a new method for task-directed exploration where actions are chosen with the goal of maximizing the future information gain about the optimal trajectory. In order to best understand the performance of this method, in this paper we have focused on the setting where we needed to make the smallest set of assumptions to apply this techniques: problems where the dynamics can be modeled by a Gaussian process and where a relatively simple trajectory optimization method suffices to find a good trajectory given GT dynamics. In these settings we attempted to fully ablate our method and tease out the contribution of the $C_\\tau^*$ cost function and its effects on the learned dynamics model compared to a wide range of methods which often used the exact same prior and planner. We believe that this is the most controlled evaluation possible for understanding the method.

One unfortunate consequence of our choice of Gaussian process dynamics is that GPs are widely known to have sample complexity exponential in the dimensionality of the domain [1] and computational complexity cubic in the number of datapoints. Across the field of RL using Gaussian processes this has restricted methods to evaluate on environments with low to moderate dimensionality [2][3][4]. Though we would have liked to evaluate our method on high-dimensional experiments such as Ant or Atari, it is not computationally practical for GP-based methods using exact computations and a sampling-based planner.

There are also quite a number of methods for scaling GPs [5] or using hybrid models with neural networks or other approaches to Bayesian dynamics modeling. Each of these requires some thought about the best way to estimate the $C_\\tau^*$ cost function given a new kind of dynamics model. As we mention in the conclusion, we are actively working on exploring the space of methods to further scale this approach, but it will take substantial trial and error, implementation effort, and methodological improvement. We hope to one day present an RL algorithm that is simultaneously state-of-the-art in sample efficiency, scalability, and robustness to misspecifications. But we believe that it is not necessary for a contribution to solve all problems at once.

We have added language to the paper making it more clear that our control problems are low-to-medium dimensional.

[1] Gaussian Processes for Machine Learning, Rasmussen and Williams, MIT Press 2006

[2] PILCO: A Model-Based and Data-Efficient Approach to Policy Search, Deisenroth & Rasmussen, ICML 2011

[3] Data-Efficient Reinforcement Learning with Probabilistic Model Predictive Control, Kamthe and Deisenroth, AISTATS 2018

[4] Sample Efficient Reinforcement Learning with Gaussian Processes, Grande et al, ICML 2014

[5] Conditioning Sparse Variational Gaussian Processes for Online Decision-making, Maddox et al, Neurips 2021

---

### Meta-Review · Area_Chair_UNWh · 2022-08-27

**Recommendation:** Accept
**Confidence:** Certain

**Metareview:**

All reviewers acknowledged to have read the rebuttal. Reviewer iWun's reply isn't visible to the authors (posted too late), see end of metareview. The most important concerns of the reviewers have been addressed by extensive replies and additional experiments. Overall the method is sound and performs well. As acknowledged by the authors, the method comes with inherent limitations to low-to-medium dimensional problems through the use of GPs. The method is useful on its own, and serves as proof-of-concept for the overall idea also for different function approximators -  but replacing GPs will require quite a bit of additional work.

***
18 Aug 2022, NeurIPS 2022 Conference Paper2167 Reviewer iWun
"Thanks to the authors for the response.

I still don't like non-strict mathematical language of the paper, but this doesn't seem to be a problem for other reviewers. In addition, I like the results of the paper, and therefore I increase my score."


**Award:**

No

---

### Decision · Program_Chairs · 2022-09-14

Accept